# Effects of Albumin on Survival after a Hepatic Encephalopathy Episode: Randomized Double-Blind Trial and Meta-Analysis

**DOI:** 10.3390/jcm10214885

**Published:** 2021-10-23

**Authors:** Meritxell Ventura-Cots, Macarena Simón-Talero, Maria Poca, Xavier Ariza, Helena Masnou, Jordi Sanchez, Elba Llop, Núria Cañete, Marta Martín-Llahí, Alberto Amador, Javier Martínez, Ana Clemente-Sanchez, Angela Puente, Maria Torrens, Edilmar Alvarado-Tapias, Laura Napoleone, Mireia Miquel-Planas, Alba Ardèvol, Meritxell Casas Rodrigo, Jose Luís Calleja, Cristina Solé, German Soriano, Joan Genescà

**Affiliations:** 1Liver Unit, Department of Internal Medicine, Hospital Universitari Vall d’Hebron, Vall d’Hebron Institut de Recerca (VHIR), Vall d’Hebron Barcelona Hospital Campus, Universitat Autònoma de Barcelona, 08035 Barcelona, Spain; mariacadaques@gmail.com (M.T.); jgenesca@vhebron.net (J.G.); 2Centro de Investigación Biomédica en Red, Enfermedades Hepáticas y Digestivas (CIBERehd), Instituto de Salud Carlos III, 28029 Madrid, Spain; Mpoca@santpau.cat (M.P.); XARIZA@clinic.cat (X.A.); jsanchezd@tauli.cat (J.S.); elballop@gmail.com (E.L.); anacs@pitt.edu (A.C.-S.); ealvaradot@santpau.cat (E.A.-T.); lnapoleone@clinic.cat (L.N.); mmiquel@tauli.cat (M.M.-P.); joseluis.calleja@uam.es (J.L.C.); csole@tauli.cat (C.S.); gsoriano@santpau.cat (G.S.); 3Department of Gastroenterology, Hospital de la Santa Creu i Sant Pau, Institut de Recerca Hospital de la Santa Creu i Sant Pau, Universitat Autònoma de Barcelona, 08041 Barcelona, Spain; 4Liver Unit, Hospital Clínic de Barcelona, 08036 Barcelona, Spain; 5Gastroenterology Department, Hospital Universitary Germans Tries i Pujol, 08916 Badalona, Spain; hmasnou.germanstrias@gencat.cat (H.M.); albaardevol@gmail.com (A.A.); 6Department of Gastroenterology, Corporació Sanitària Parc Taulí, 08208 Sabadell, Spain; mcasasr@tauli.cat; 7Hospital Universitario Puerta de Hierro, Puerta de Hierro Health Research Institute (IDIPHIM), 28222 Madrid, Spain; 8Liver Section, Gastroenterology Department, Hospital del Mar, Hospital del Mar Medical Research Institute (IMIM), Universitat Autònoma de Barcelona, 08003 Barcelona, Spain; Ncanete@parcdesalutmar.cat; 9Gastrointestinal Department, Hospital Moisès Broggi, 08970 Sant Joan Despí, Spain; martinllahi@gmail.com; 10Hepatology Unit, Gastroenterology Department, Hospital de Bellvitge, ICS, Research Group of Hepato-biliary and Pancreatic Diseases (IDIBELL), Barcelona University, 08907 L’Hospitalet de Llobregat, Spain; aamadornavarrete@gmail.com; 11Deparment of Gastroenterology, Hospital Universitario Ramon y Cajal, 28034 Madrid, Spain; martinez.gonzalez.javier@gmail.com; 12Liver Unit, Digestive Department, Hospital General Universitario Gregorio Marañón, Complutense University of Madrid, 28007 Madrid, Spain; 13Digestive Disease Department, University Hospital Marqués de Valdecilla, Instituto de Investigación Sanitaria Valdecilla (IDIVAL), 39008 Santander, Spain; angelapuente@hotmail.com; 14Department of Medicine, Universitat de Vic-Universitat Central de Catalunya (UVIc-UCC), 08500 Vic, Spain

**Keywords:** clinical trial, meta-analysis, albumin, hepatic encephalopathy

## Abstract

No therapies have been proven to increase survival after a hepatic encephalopathy (HE) episode. We hypothesize that two doses of albumin could improve 90-day survival rates after a HE episode. Methods: (1) A randomized double-blind, placebo-controlled trial (BETA) was conducted in 12 hospitals. The effect of albumin (1.5 g/kg at baseline and 1 g/kg on day 3) on 90-day survival rates after a HE episode grade II or higher was evaluated. (2) A meta-analysis of individual patient’s data for survival including two clinical trials (BETA and ALFAE) was performed. Results: In total, 82 patients were included. Albumin failed to increase the 90-day transplant-free survival (91.9% vs. 80.5%, *p* = 0.3). A competing risk analysis was performed, observing a 90-day cumulative incidence of death of 9% in the albumin group vs. 20% in the placebo (*p* = 0.1). The meta-analysis showed a benefit in the albumin group, with a lower rate of clinical events (death or liver transplant) than patients in the placebo (HR, 0.44; 95% CI, 0.21–0.82), when analyzed by a competing risk analysis (90-days mortality rate of 11% in the albumin group vs. 30% in the placebo, *p* = 0.02). Conclusions: Repeated doses of albumin might be beneficial for patient’s survival as an add-on therapy after an HE episode, but an adequately powered trial is needed.

## 1. Introduction

Hepatic encephalopathy (HE) is a life-threatening complication of cirrhosis. HE has 90-day mortality rates of 20% in patients with grade II and 45% in patients with grade III or IV [1]. Despite high mortality rates, therapeutic options are limited, and since the introduction of rifaximin 10 years ago, a drug used for secondary prophylaxis, no other significant advances have been made [2]. The current HE treatment is based on nonabsorbable disaccharides and nonabsorbable antibiotics. No treatment has been proven to increase survival rates among patients with a HE episode [3]. Albumin is a multifunctional protein with a unique and complex structure. It is the main intravascular volume regulator with homeostatic and transportation functions, also presenting immunomodulator, antioxidant, and detoxification effects [4]. Albumin is the standard of care for some liver-related decompensations, reducing the incidence of renal impairment in spontaneous bacterial peritonitis (SBP) [5,6] and preventing renal failure after massive paracentesis [7]; it is also used as an add-on therapy for hepatorenal syndrome (HRS) [8]. Recently, albumin failed to demonstrate efficacy in the prevention of complications of cirrhosis (HE, gastrointestinal bleeding, hyponatremia, renal failure, and bacterial infections) in patients on the liver-transplant waiting list [9]. In contrast, two double-blind, randomized clinical trials pointed to a possible effect of albumin on survival rates [10,11]. The results of the ALFAE study [11] were especially encouraging since the albumin group exhibited a decrease of 29% in mortality rate. Importantly, none of these two studies were designed to address survival. Another recent study failed to demonstrate an effect of albumin to increase in-hospital survival, although a lower frequency of infections and a higher survival rate were observed within the treatment group [12]; in a similar manner, another randomized trial also failed to prevent decompensation (i.e., infections, kidney disfunction) or death among hospitalized cirrhotic patients [13]. Finally, the ANSWER study proved that long-term albumin administration prolongs overall 18-month survival rates in patients with decompensated cirrhosis. Approximately 25% of patients in each arm presented with previous HE history, and the cumulative incidence of HE was drastically reduced in the albumin group [14]. In the present study, we aimed at investigating whether albumin could increase 90-day survival rates after an acute HE episode.

## 2. Materials and Methods

### 2.1. Albumin Infusion Effect in Hepatic Encephalopathy (BETA)

The study was a double-blind, placebo-controlled, multicenter randomized trial designed to evaluate the effect of albumin on mortality in cirrhotic patients with an acute episode of HE. Consecutive patients recruited from June 2015 to April 2019 admitted to 6 tertiary Spanish hospitals at the beginning, and later 14 tertiary hospitals were included (Appendix A). A block size of 10 was used to randomize patients 1:1, who were also stratified by creatinine (<1.2 mg/dL or ≥1.2 mg/dL) and center. The randomization sequence was generated by the pharmacy epartment of Hospital Vall d’Hebron. The study protocol conformed to the ethical guidelines of the 1975 Declaration of Helsinki as reflected in a priori approval by the appropriate institutional review committee. Informed consent in writing was obtained from each of the subjects prior to enrollment or by an authorized relative when the patient presented with impaired mental status; the patient was asked to re-consent once recovered from the HE episode. This study is a registered trial (ClinicalTrials.gov identifier: NCT02401490; EudraCT number: 2014-004809-33).

### 2.2. Patient Selection

Subjects were eligible for inclusion if they met the following criteria: aged between 18 and 85 years; had a diagnosis of cirrhosis as defined by clinical, laboratory, or radiological findings; and experienced a HE episode grade II or higher assessed by the West Haven scale within 48 h prior to study inclusion (HE and its grade were evaluated at the hospital and registered in the medical record). A consent form was required. The exclusion criteria were as follows: terminal illness, neurological comorbidities that impaired mental status, active gastrointestinal bleeding during the previous 48 h, need of mechanical ventilation, hemodialysis or vasoactive support, model for end-stage liver disease (MELD) score lower than 15 or higher than 25 at the time of inclusion, clinical conditions that contraindicate the administration of albumin or previous hypersensitivity to albumin, any medical condition that had or would require albumin administration during a period of 7 days prior to or after inclusion, and the presence of an acute-on-chronic liver failure as defined by the presence of one severe single organ failure (except HE) or any multiorgan failure (except HE) conferring more than 15% mortality at 28 days [15]. In premenopausal women, pregnancy and breastfeeding were exclusion criteria. Amendments to the protocol changing the MELD score range from 15 to 25 to 14 to 30 were approved in order to increase recruitment.

### 2.3. Study Protocol

The study medication was prepared in 100 mL flasks (20 g of albumin or 100 mL of 0.9% saline solution) by the pharmacy department and blindly delivered to the study investigators. The first dose was administered within the first 48 h of admission with a HE grade ≥ 2 regardless of the resolution at the time of infusion (1.5 g/kg of albumin or the equivalent mL of saline solution). The second dose (1 g/kg of albumin or the equivalent mL of saline solution) was administered 48 to 72 h after the first administration. Both doses were adjusted according to ideal weight (without ascites) and infused at 5 mL/min. All patients were assessed for HE-precipitating factors. The underlying causes were properly addressed according to guidelines [3]. Moreover, laxative treatment was administered. At the conclusion of the episode, all the patients were placed on rifaximin (600 mg/12 h) and non-absorbable disaccharides and recommended a normal protein diet and hydration. Visits were performed on screening, day 0, day 2–3, and day 7 according to the protocol (Appendix A).

### 2.4. Outcome

The primary outcome was to evaluate the effect of albumin administration 90-day transplant-free survival after a grade II HE episode, based on results obtained from other studies [11]. Due to the competing risk of transplantation on mortality, 90-day survival with transplantation as a competing event was preferred. Secondary outcomes included the following: to evaluate the effect of albumin on 28- and 180-day survival, decrease hospital admissions due to HE or any other liver-related complications, and evaluate the efficacy of albumin on avoiding HE relapse during all follow-up periods.

### 2.5. Statistics

The sample size was set at 116 patients (58 patients per group) to detect 25% differences on mortality with an α error of 0.05 and beta error of 0.80. According to previous studies, mortality within the placebo group was estimated to be around 44% [11,15]. The final sample size (*n* = 128) was calculated according to a potential loss of 10% of patients. At the end of the randomization period, we had 82 patients representing 64% of the sample size. The study was terminated due to low enrollment and excessive duration. Results are presented as frequencies and percentages for categorical variables, means and SDs for normal continuous variables, and median, quartile 1, and quartile 3 for not normal continuous variables. Univariate analyses, using Chi-square, Student’s *t* test, and Mann–Whitney U test were carried out to compare variables between the treatment and the placebo group. To evaluate the evolution of clinical outcome frequencies (i.e., ascites, HE, minimal hepatic encephalopathy (MHE), and infection), and during the follow-up period, the Skilling–Mack test was used. The main outcome was assessed as a combined variable, mortality or liver transplant, using Kaplan–Meier curves and the log-rank Mantel–Cox test. As explained, overall survival analysis was also performed by using a sub-hazards model for competing risks. This model was chosen in order to account for liver transplantation as a ‘‘competing’’ event for mortality, based on the consideration that transplantation at a given time clearly modifies the probability of dying for a specific patient at each subsequent time point, especially considering the short follow-up of the study. For all analyses, type I error was set at <0.05. All authors had access to the study data to review and approve the final manuscript.

### 2.6. Metanalysis: Albumin Infusion Effect in Hepatic Encephalopathy (BETA) and Effects of Intravenous Albumin in Patients with Cirrhosis and Episodic Hepatic Encephalopathy (ALFAE)

Exclusively for a survival analysis, and in order to reach the estimated sample size, we incorporated the patients from the ALFAE study in the analysis. The characteristics of the ALFAE study have been extensively explained elsewhere [11]. In summary, it was a randomized, double-blind, placebo-controlled multicenter study performed in four tertiary Spanish centers that included 56 patients (albumin group *n* = 26 and saline group *n* = 30). Patients were randomized to albumin or placebo. Albumin was administered intravenously at a dose of 1.5 g/kg at inclusion (day 1) and 1.0 g/kg after 48 h (day 3), and adjusted by ideal weight. Saline (NaCl 0.9%, Grifols, SA, Barcelona, Spain) was administered at equivalent volumes. Treatment was infused at a rate of 5 mL/min.

The inclusion and exclusion criteria as well as an extensive comparison between patients from both studies can be found in the Appendix A.

### 2.7. Statistics

A metanalysis was performed by calculating hazard ratios (HRs) and 95% CIs. Because slight differences in the patient’s enrollment may have had an impact on the magnitude of the effect, we chose to pool data and compare it using a random effects model. Type I error was also considered to be at 5%. Statistical heterogeneity was calculated by the I-square. Values less than 30% were considered as indicators of low heterogeneity. Survival analysis was also performed by using a sub-hazards model for competing risks. Finally, to assess the presence of independent mortality risk factors, we used a method based on the improved likelihood ratio (*p*-value less than 0.1) and the Akaike information criterion. All analyses were performed using the software Stata 15.1 (StataCorp. 2017. *Stata 15 Base Reference Manual*. College Station, TX, USA: Stata Press).

## 3. Results

Overall, 677 patients were assessed for eligibility, of whom 82 were randomized between June 2015 and April 2019 and followed for 180 days. The most frequent cause of exclusion was MELD restriction (MELD score lower than 15 or higher than 39) alone or in combination with other excluding criteria (Figure 1). In total, 40 patients were allocated to the albumin group and 42 to the placebo group. One patient from the placebo group withdrew consent before finalizing the first day infusion. All patients reached the second albumin infusion, except for the one who withdrew before the first dose.

### 3.1. Basal Clinical Characteristics

Approximately 67% of the patients were males with a medium age of 68.6 and with previous episodes of liver-related decompensations (Table 1). The medium MELD score was 17 in both groups. The precipitating factors were distributed homogenously across both groups, except for a higher percentage of diuretic use among the treatment group. Most of the patients presented with an HE episode grade II assessed by the West Haven scale at admission. At the time of infusion (day 1 and day 3), the grade of HE did not differ significantly between the two groups. The most common treatment for HE in both groups were non-absorbable disaccharides.

### 3.2. Complications during Follow-Up

In total, 13 patients (31%) from the placebo group and 18 (45%) from the albumin group presented HE episodes that did not require hospitalization during the follow-up period. Furthermore, 18 patients from the placebo group and 11 from the albumin group were admitted at least on one occasion due to a HE episode. Patients from the placebo group had a mean of 1.72 HE-related admissions, while a mean of 1.55 admissions were recorded within the albumin group. Admissions related to other liver-related complications did not differ between groups. Patients free of overt HE were evaluated for MHE during the follow-up with the psychometric hepatic encephalopathy score (PHES) score. During follow-up, 25 patients from the albumin group and 16 from the placebo group had at least one PHES score. Interestingly, at any time point (except 60 days) the prevalence of MHE among the albumin group was lower compared to the placebo group, although these differences did not reach statistical significance (Table 2).

No differences on the incidences of infections and ascites were observed between groups during the follow-up.

### 3.3. Survival

The transplant-free survival for the whole cohort was 83.95% at 90 days. Patients from the albumin group presented higher 90-day transplant-free survival when compared to the placebo group, with percentages of 87.36% vs. 80.49%, respectively, but no statistical significance was reached (*p* = 0.38). Both death and transplant were considered competing events. Therefore, we performed a secondary competing risk analysis observing a 90-day cumulative incidence of death of 9% in the albumin group vs. 20% in the placebo group (*p* = 0.1) (Figure 2).

During the 180-day follow-up, the transplant-free survival was also higher among the albumin group when compared to the placebo group (79.7% vs. 67.8%, respectively, *p* = 0.2). The 180-day cumulative incidence of death was 11% among the albumin group and 28% in the placebo group, again without reaching statistical significance (*p* = 0.09) (Appendix A).

### 3.4. Safety and Tolerability

In terms of adverse events, 36 patients from the albumin group and 37 from the placebo group reported at least one adverse event. Moreover, 185 adverse events were reported by patients from the albumin group and 184 by patients from the placebo group. The most frequents AEs were related to the underlying liver disease and infections (Appendix A). More than 80% of the AEs in both groups required treatment. Of those, 47 AEs from the albumin group and 48 from the placebo group were considered to be severe (Appendix A). Up to 60% of AEs required hospitalization. In total, 6 SAEs from the albumin group and 14 from the placebo group were linked to the death of the patient. Importantly, none of the AEs or SAEs in the albumin group were considered to be related with the study treatment.

### 3.5. Meta-Analysis Results: BETA and ALFAE Studies

Next, we performed an individual patient’s data meta-analysis of both trials. Overall, the two studies included 138 patients, with 66 patients in the albumin group and 72 patients in the treatment group (Appendix A). We observed a slightly higher rate of infection among the ALFAE patients; we therefore analyzed whether this fact might have impacted mortality within the ALFAE group. The distribution of deaths and transplants during the first 15 days after randomization was similar within all the groups, pointing to a limited effect of baseline infections on mortality and liver transplant (Appendix A).

Subgroup difference testing showed no significant differences between both studies when looking at treatment response (heterogeneity χ^2^ = 0.67; d.f. = 1; *p* = 0.4, I2 = 0%, and Tau-square = 0). Patients who received treatment with albumin had a significantly lower rate of clinical events (death or liver transplant) than patients in the placebo group (HR, 0.44; 95% CI, 0.21–0.82) (Figure 3A). The overall 90-day transplant-free survival was 84.2% in the albumin group and 66.2% in the placebo group, *p* = 0.018 (Figure 3B). To keep consistency, we then performed a competing risk analysis with transplant as a competing event. The cumulative incidence of death within the albumin group was 11%, while it reached 30% in the placebo group, *p* = 0.02 (Figure 3C). The independent risk factors of mortality at 90 days for the joint sub-analysis were albumin treatment, EH severity at baseline, MELD score, and baseline sodium (Appendix A).

## 4. Discussion

In the present clinical trial, the efficacy of albumin in increasing 90-day survival after a HE episode was evaluated. Unfortunately, the efficacy of albumin could not be demonstrated based on the results of the trial. The study only reached 64% of the estimated sample size, and mortality within the placebo group was half the expected rate. The combination of these facts has been critical in the outcome of the study. Despite these drawbacks, a tendency towards a better survival outcome was observed within the albumin group. To address the methodologic question, an individual patient meta-analysis of two clinical trials (BETA and ALFAE) to analyze survival was performed [11]. Both trials were conducted by our group in patients with an acute HE episode. The inclusion and exclusion criteria, as well as the therapeutic study protocol from both studies, are almost identical. The only difference between the ALFAE and the BETA studies was the lack of MELD restrictions in the ALFAE study, as well as the definition of ACLF that has changed over the years. An extensive comparison between groups was performed, and the most relevant differences were a higher incidence of infection as a HE precipitant event within the ALFAE group as well as worse analytical and severity scores among the BETA group. Remarkably, no differences between MELD scores were found. Thus, we analyzed whether the higher incidence of baseline infections might have impacted the mortality rate within the ALFAE group. The distribution of deaths and transplants during the first 15 days after randomization was similar within all the groups, pointing to a limited effect of baseline infections on mortality. When we analyzed both studies together, the protective effect of albumin was evident. The overall 90-day transplant-free survival was 84.2% in the albumin group and 66.2% in the placebo group. We also performed a competing risk analysis in order to account for liver transplant as an event ‘‘competing’’ with mortality, with a cumulative incidence of death within the placebo group triplicating the one from the albumin group (30% vs. 11%). The meta-analysis results support this finding, pointing to the sample size as a critical flaw of the BETA study. Besides the albumin administration, baseline sodium, MELD score, and baseline HE severity were identified as 90-day mortality risk factors, consistent with risk factors already reported in other studies [1].

The effects of albumin on other clinical outcomes were also evaluated. In the current studies, no effect on the number of HE episodes, hospitalizations, and infections was observed. Interestingly, a recent clinical trial with long-term albumin administration has proven to increase long-term survival and reduce the cumulative incidence of HE during follow-up among cirrhotic patients [14]. Thus, repeated albumin administration, even at low doses, during prolonged periods of time might be the critical point to achieve a clinical benefit on survival. We hypothesize that the beneficial effect of albumin in our cohort might be due to its effect on systemic inflammation, as well as in portal hemodynamics [4,16].

Another critical question is to define who would benefit from this therapy. In our study, we established some MELD cut-offs, as well as the exclusion of patients with ACLF that carry out a 28-day mortality higher than 15%. On the contrary, the ANSWER study, although not establishing any explicit MELD restriction, only allowed the inclusion of decompensated cirrhotic patients. Therefore, the median baseline MELD score of the ANSWER patients was between 3 and 4 points lower than the MELD score of our patients. The fact that 80% of the patients from the ANSWER study did not present any HE episode before randomization might play a key role in the lower incidence of HE during follow-up, since the presence of previous HE episodes is one of the most important risk factors to develop a new episode [1].

Although our study points to a significant increase in 90-day survival among patients who received albumin, we fully acknowledge the methodological flaws including having two clinical trials that did not reach the target number of enrolled patients, with slight differences in inclusion criteria (i.e., MELD restriction in the BETA study and bilirubin restriction in the ALFAE study), end-points, and the need to merge them to see the expected effect. However, the results of the meta-analysis are also clear, and differences between both trials were taken into account, and they at least indicate that this matter deserves further studies. Therefore, a larger clinical trial is needed to confirm our initial hypothesis.

## Figures and Tables

**Figure 1 jcm-10-04885-f001:**
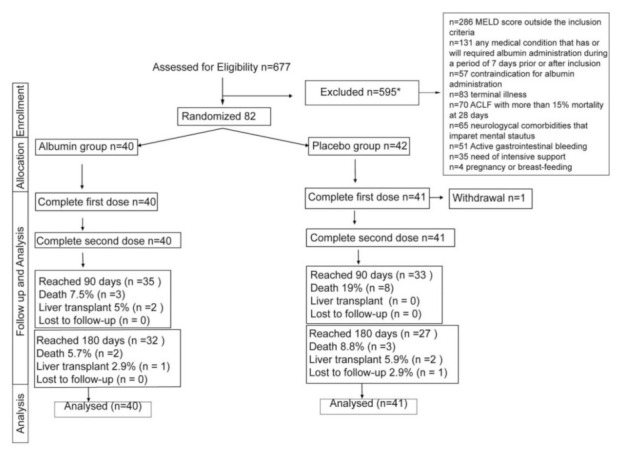
Study flow chart. * Patients could have more than one exclusion criteria. In 20 patients, the exclusion criteria were not recorded.

**Figure 2 jcm-10-04885-f002:**
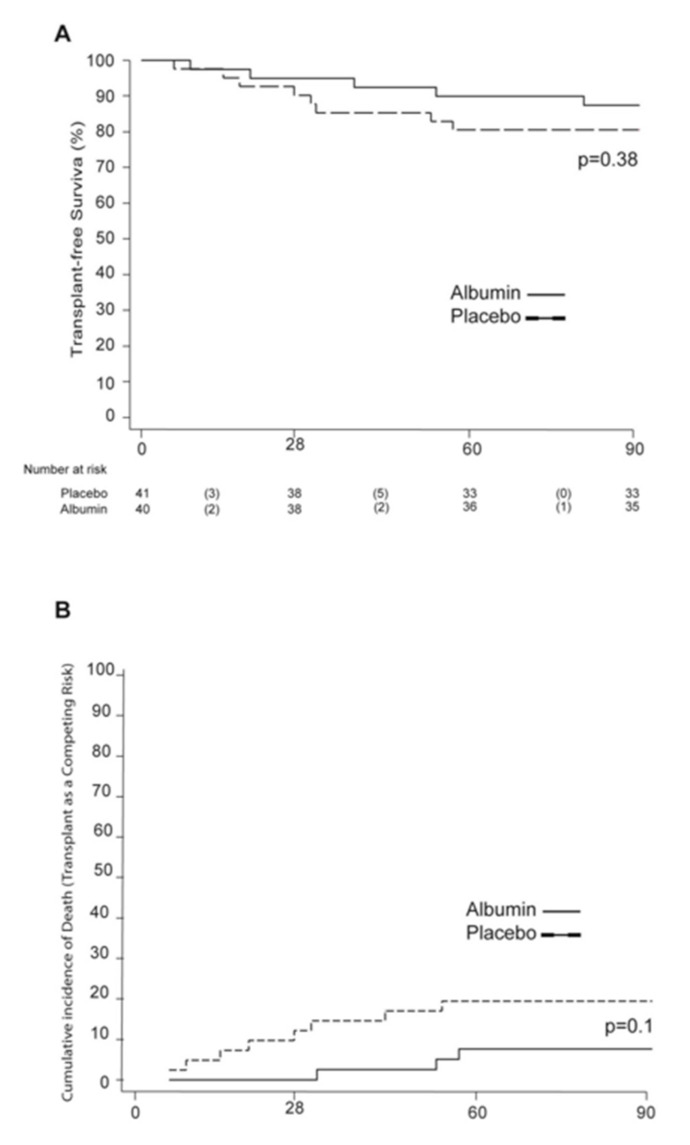
Transplant-free survival and cumulative incidence of death with transplant as a competing risk. (**A**) Kaplan–Meier estimates of transplant-free survival at 90 days. (**B**) The 90-day cumulative incidence of death with transplant as a competing risk.

**Figure 3 jcm-10-04885-f003:**
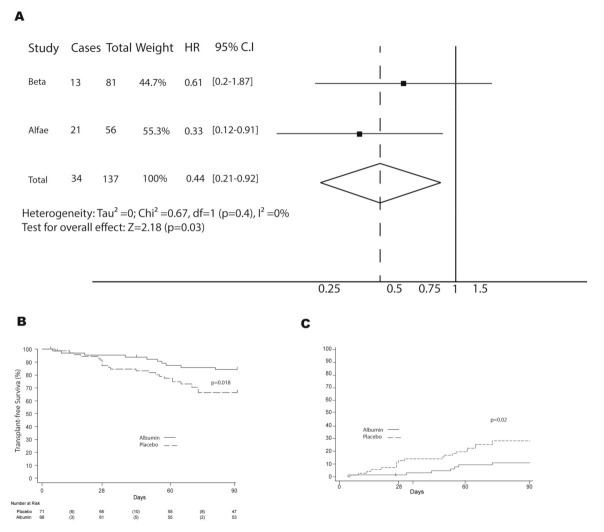
Forest plot of transplant-free survival and cumulative incidence of death with transplant as a competing risk for BETA and ALFAE studies. (**A**) Forest plot comparison of transplant-free survival; (**B**) Kaplan–Meier estimates of transplant-free survival at 90 days. (**C**) The 90–day cumulative incidence of death with transplant as a competing risk. Abbreviations: hazard ratio, HR; confidence interval, CI.

**Table 1 jcm-10-04885-t001:** Baseline characteristics of the BETA trial by treatment group.

	Placebo Group	Albumin Group
	*n* = 42	*n* = 40
**Male**, *n* (%)	26(61.9)	29(72.5)
**Age**, median (IQR)	69.1(63.3–75.3)	66.5(59.9–73.6)
**BMI**, median (IQR)	24.5(23.2–26.5)	24.2(23.2–26.8)
**Etiology**, *n* (%)		
Alcohol related	22(57.7)	19(47.5)
MAFLD	2(4.9)	4(10)
Alcohol and HCV	4(9.8)	6(15)
HCV	5(12.2)	6(15)
Alcohol and MAFL	2(4.9)	--
Other *	7(8.8)	5(6.1)
**Previous decompensations**, *n* (%)		
Ascites	35(83.3)	30(75)
Hepatic encephalopathy	27(64.3)	26(65)
Spontaneous bacterial peritonitis	7(16.7)	7(17.5)
Gastrointestinal bleeding	11(26.2)	11(27.6)
Hepatorenal syndrome	2(4.8)	3(7.5)
Hepatocellular carcinoma	5(11.9)	4(10)
**Other comorbidities**, *n* (%)		
Hypertension	26(66.7)	23(59)
Dyslipidemia	9(23.1)	7(17.9)
Diabetes	20(51.3)	14(35.9)
**Severity score**, median (IQR)		
MELD	17(16–20)	17(15–20)
**Treatment at inclusion**, *n* (%)		
Rifaximin and/or lactulose	35(83.3)	31(77.5)
**Laboratory parameters**, median (IQR)		
Hemoglobin g/dL	10.6(9.9–13)	10.9(9.3–11.8)
Leukocytes x109/L	5.06(3.8–7.4)	5.09(4.06–6.6)
Platelets x10E9/L	82.5(64–109)	76.5(57.5–106.5)
Sodium mEq/L	135.8(132.8–138.6)	136.2(134–139)
AST IU/L	44(31–64)	51(36–75)
ALT IU/L	26(18–35)	32.5(22–43.5)
Bilirubin mg/dL	3.2(1.7–4.6)	2.97(1.91–5)
Albumin g/dL	2.85(2.35–3.01)	2.6(2.41–2.93)
INR	1.53(1.37–1.71)	1.55(1.35–1.83)
Creatinine (mg/dL)	1.08(0.77–1.53)	0.99(0.7–1.32)
**Mean Arterial Pressure (mmHg)**	82.6(76–94.3)	78.7(72.5–89.3)
**Current liver-related decompensations**, *n* (%) ♦		
Ascites	19(45.2)	17(42.5)
Hepatorenal syndrome	1(2.4)	--
**Precipitant factors**, *n* (%)		
Infection	11(26.1)	12(30)
Constipation	9(21.4)	9(22.5)
Dehydration	4(9.5)	3(7.5)
Diuretics	14(33.3)	23(59) **
**West Haven at screening**, *n* (%)		
II	30(71.4)	31(77.5)
III	11(26.2)	9(22.5)
IV	1(2.4)	--
CHESS scale	2.5 (1–6)	2(1–4)

* Other etiologies included hepatitis B and delta virus, primary biliary cholangitis, primary sclerosing cholangitis, secondary cholangitis, cryptogenic, Wilson’s disease, and autoimmune hepatitis. ** *p* = 0.02. ♦ Ascites did not require large-volume paracentesis and albumin infusion at the time of inclusion, and HRS was diagnosed within hours following inclusion. Body mass index, BMI; metabolic-associated fatty liver disease, MAFLD; hepatitis C virus, HCV; MELD model of end-stage liver disease, MELD; aspartat aminotrasferase, AST; alanine aminotransferase, ALT; hepatic encephalopathy, HE; clinical hepatic encephalopathy staging scale, CHESS.

**Table 2 jcm-10-04885-t002:** Baseline characteristics of the BETA trial by treatment group.

Time Point	Placebo Number of Patients with MHE/Number of Total Evaluated Patients and Percentage	Albumin Number of Patients with MHE/Number of Total Evaluated Patients and Percentage
Day 2–3	6/9 (66.6)	5/13(38.5)
Day 7	7/12(58.3)	4/17(23.5)
Day 30	4/10(40)	5/16(31.3)
Day 60	1/4(25)	5/14(35.7)
Day 90	6/12(50)	4/16(25)
Day 180	2/6(33.3)	3/13(7.7)

No significant differences were found at any time point. Abbreviations: minimal hepaticencephalopathy, MHE.

## Data Availability

Due to privacy and ethical concerns, neither the data nor the source of the data can be made available.

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
