# Peer review of "Effects of Albumin on Survival after a Hepatic Encephalopathy Episode: Randomized Double-Blind Trial and Meta-Analysis"

_jcm, 2021, doi:10.3390/jcm10214885_

Round 1

Reviewer 1 Report

Thank you for asking me to review this manuscript. Ventura-Cots, Simon-Talero et al conducted an RCT of 82 patients admitted with hepatic encephalopathy (BETA study), and also conducted a meta-analysis of on this topic (between BETA and previously-published ALFAE). Notable inclusion criteria were MELD 15-25, any condition for which albumin would be indicated +/- 7 days of inclusion, and ACLF; other than the MELD criterion, these criteria overlapped with previously-published albumin studies that the authors led. The study did not reach its target enrollment. There was no significant difference in transplant-free survival or death without transplant in the BETA study. In the meta-analysis, albumin was associated with decreased transplant-free survival and decreased death without transplant.

The methodology is solid and limitations are acknowledged by the authors.

Comments:

  1. While 677 patients were screened, only 82 were included. The most common reason for exclusion was MELD not within the specified range. Was this mostly because their MELD was too high or too low? Could the authors comment on why they included it (whereas their earlier albumin studies did not require this), and what effects that may have had on the results?
  2. I was surprised by how many patients had to be excluded based on having a condition requiring albumin recently. What were these conditions? Large volume paracenteses mostly I assume?
  3. Why did the BETA study not reach goal sample size? Too many patients excluded, difficulty enrolling?

Author Response

October 20th, 2021

We would like to thank the Journal of Clinical Medicine for allowing us to resubmit a revised version of our manuscript “Effects of albumin on survival after a hepatic encephalopathy episode: randomized double-blind trial and meta-analysis”. Please find the point-by-point answer to the comments from the reviewers as well the revised version of our manuscript. We hope that you will now find the revised manuscript suitable for publication in the Journal of Clinical Medicine.

Sincerely,

Meritxell Ventura-Cots, MD, Ph.D

Macarena Simón-Talero, MD, Ph.D

Liver Unit, Department of Internal Medicine

Hospital Universitari Vall d’Hebron

Reviewer #1

We would like to thank the reviewer for the comments.

Point 1 While 677 patients were screened, only 82 were included. The most common reason for exclusion was MELD not within the specified range. Was this mostly because their MELD was too high or too low? Could the authors comment on why they included it (whereas their earlier albumin studies did not require this), and what effects that may have had on the results?

Response 1

The reviewer raises a very relevant point, the high number of screened patients. Certainly, the main reason for exclusion was to present with a model for end-stage liver (MELD) score outside the inclusion criteria. Unfortunately, we will not be able to provide the exact number of screen patients with MELD score above or below the range limit, since our electronic case report form only recorded the variable “MELD range outside the inclusion criteria”. We might be able to provide the data after a case by case reviewer chart; nevertheless, we shall need more time to do it, since up to 14 centers were involved.

The range 15-25 was chosen based on two principles 1) The lower range was established to exclude those patients with hepatic encephalopathy (HE) associated with an excellent vital prognosis; this type of HE is related to a low MELD score (i.e. patients with the presence of porto-systemic shunts and absence of liver disease or associated with a minimal alteration of liver function) (Laleman, W, et al. Embolization of large spontaneous portosystemic shunts for refractory hepatic encehalpathy: a multicenter survey on safety and efficacy. Hepatology, 2013). We hypothesized that those patients would not benefit from albumin infusion, and 2) The higher range was established to exclude patients with extremely high mortality or liver transplant probability in whom albumin infusion benefit was thought to be marginal.

As reflected in the manuscript, amendments to the protocol on both the lower and the upper cut-off were changed to 14-30 in order to increase recruitment and at the same time maintaining the rational for MELD restriction (page 4).

In the ALFAE study although MELD score was not an exclusion criteria, a restriction base on bilirubin (i.e >5 mg/dL) was applied. As shown in supplementary table 2 the different exclusion criteria did not have a repercussion on the patient’s baseline characteristics including the medium MELD score; for this reason, no differences were expected.

A comment has been incorporated in the discussion section (page 11).

Point 2 I was surprised by how many patients had to be excluded based on having a condition requiring albumin recently. What were these conditions? Large volume paracenteses mostly I assume?

Response 2. The reviewer raises a very interesting point, the large number of patients requiring albumin. Twenty-two percent of the screened patients fell under the next exclusion criteria “has or will required albumin administration during a period of 7 days prior or after inclusion”. As the reviewer points out, the main reason for albumin administration was the requirement of large volume paracentesis. Some of the patients were already under a large volume paracentesis program, while others required the procedure during the current decompensation episode. A lower percentage of patients presented with a suspicion of hepatorenal syndrome and/or spontaneous bacterial peritonitis. We would like to point out that in real life practice, patients excluded from our study because of recent or periodic albumin administration might also benefit from albumin infusion for HE; we decided to exclude them for the clinical trial to have a more homogenous population.

Point 3 Why did the BETA study not reach goal sample size? Too many patients excluded, difficulty enrolling?

Response 3. As the reviewer points out the main reasons for not achieving the sample size were the difficulty in enrolling, as well as the large number of patients excluded.

Reviewer 2 Report

Ventura-Cots et al. conducted a randomized placebo-controlled study to examine the effects of albumin infusion (1.5g/kg at baseline and 1g/kg on day 3) on outcome among patients with cirrhosis following an OHE event. Because the study failed to meet its enrollment target, the individual patient results were combined with those from a previously published Spanish study (Simon-Talero et al., 2013) which was similar, although not identical in design. While the outcome of the newly reported study failed to show a significant difference in survival, albeit directionally favorable difference in favor of the albumin group, combining the data from the two studies resulted in a statistically significant survival difference in favor of the albumin group.

The conclusions of the study are seriously flawed, a point which the authors fully acknowledge. The authors also point out that the results need to be verified or rejected by future studies. Despite these shortcomings, the manuscript reports a lot of hard-won real world data, as outlined in my comments to authors.

General

  • As the authors point out, the upper MELD exclusion in the present study was 25 and the overall survival ~84%. There was no upper MELD limit in the earlier Simon-Talero study, which therefore included sicker subjects. Instead of presenting only the results of the present study and two studies combined, it would be helpful for the authors to elaborate on how out differed in the two studies.
  • If albumin does improve survival, it is not clear how, as it no effect on the number of HE episodes, hospitalizations and infections. The authors should elaborate on how albumin renders its purported benefit and whether there were any differences in the risk or rate of potential lethal complications.
  • It would be helpful for the authors to comment briefly on the design of a future study to verify or refute the current results.

Minor

  • Why was the upper MELD exclusion set to 25? This is arguably low and accounted for ~50% of the patients excluded.
  • There are errors in English grammar or syntax in the manuscript.

Author Response

October 20th, 2021

We would like to thank the Journal of Clinical Medicine for allowing us to resubmit a revised version of our manuscript “Effects of albumin on survival after a hepatic encephalopathy episode: randomized double-blind trial and meta-analysis”. Please find the point-by-point answer to the comments from the reviewers as well as the revised version of our manuscript. We hope that you will find the revised manuscript suitable for publication in the Journal of Clinical Medicine.

Sincerely,

Meritxell Ventura-Cots, MD, Ph.D

Macarena Simón-Talero, MD, Ph.D

Liver Unit, Department of Internal Medicine

Hospital Universitari Vall d’Hebron

Reviewer #2

We would like to thank the reviewer for the comments.

Point 1 As the authors point out, the upper MELD exclusion in the present study was 25 and the overall survival ~84%. There was no upper MELD limit in the earlier, which therefore included sicker subjects. Instead of presenting only the results of the present study and two studies combined, it would be helpful for the authors to elaborate on how out differed in the two studies.

Response 1. The reviewer raises a very relevant point, the differences between both studies. The inclusion and exclusion criteria, as well as an extensive comparison between patients from both studies, were depicted in the Supplementary methods and Supplementary Table 2. Albeit, Simon-Talero’s study did not exclude patients according to MELD score, a bilirubin (i.e >5 mg/dL) restriction was applied. This criterion allowed to exclude severely ill patients. As showed in supplementary table 2, the different exclusion criteria did not have a repercussion on patient baseline characteristics.

A comment on the discussion section has been added (page 10-11, first and fourth paragraph).

Point 2 If albumin does improve survival, it is not clear how, as it has no effect on the number of HE episodes, hospitalizations and infections. The authors should elaborate on how albumin renders its purported benefit and whether there were any differences in the risk or rate of potential lethal complications.

Response 2. We fully agree with the reviewer comment. Since no clear reduction on the number of HE episodes, hospitalizations and infections were seen, we hypothesize that albumin must have other beneficial effects on our cohort of patients. Several studies support the hypothesis that albumin has an effect on systemic inflammation, as well as on portal hemodynamics (Garcia-Martinez R et al.  Albumin: pathophysiologic basis of its role in the treatment of cirrhosis and its complications.  Hepatology. 2013 and Fernández J, et al. Effects of Albumin Treatment on Systemic and Portal Hemodynamics and Systemic Inflammation in Patients With Decompensated Cirrhosis.  Gastroenterology. 2019).

We have incorporated a comment on the discussion section (page 11).

Point 3 It would be helpful for the authors to comment briefly on the design of a future study to verify or refute the current results.

Response 3. We appreciate this valuable comment. Future studies should be focused on hard aims such as survival and hospitalizations, as well as improving the quality of life, and include a wider spectrum of patients focusing on real life clinical practice without MELD score restrictions.

Point 4 Why was the upper MELD exclusion set to 25? This is arguably low and accounted for ~50% of the patients excluded.

Response 4. The main reason for exclusion was to present with a model for end-stage liver score (MELD) outside the inclusion criteria (14-25). Certainly, approximately 50% of patients were excluded based on this criterion, nevertheless some of those patients were excluded because of a low MELD score. Based on the Spanish liver transplant list we initially decided to set up the upper MELD limit at 25, expecting a very short period of time in the transplant waiting list for these patients. Further on, to better reflect the general population and to increase the recruitment rate, we increased the upper limit of MELD to 30.

Point 5 There are errors in English grammar or syntax in the manuscript.

Response 5. We appreciate this comment. We have made changes accordingly.

Round 2

Reviewer 2 Report

I appreciate the author's response to my queries